# On the Pareto Front of Multilingual Neural Machine Translation

**Liang Chen**[1][*]  **Shuming Ma**[2][*]  **Dongdong Zhang**[2]  **Furu Wei**[2]  **Baobao Chang**[1][†]

[1]National Key Laboratory for Multimedia Information Processing,
School of Computer Science, Peking University
[2]Microsoft Research
`leo.liang.chen@outlook.com`  `chbb@pku.edu.cn`
`{shumma,dozhang,fuwei}@microsoft.com`

## Abstract

In this work, we study how the performance of a given direction changes with its sampling ratio in Multilingual Neural Machine Translation (MNMT). By training over 200 multilingual models with various model sizes, data sizes, and language directions, we find it interesting that the performance of certain translation direction does not always improve with the increase of its weight in the multi-task optimization objective. Accordingly, scalarization method leads to a multitask trade-off front that deviates from the traditional Pareto front when there exists data imbalance in the training corpus, which poses a great challenge to improve the overall performance of all directions. Based on our observations, we propose the Double Power Law to predict the unique performance trade-off front in MNMT, which is robust across various languages, data adequacy, and the number of tasks. Finally, we formulate the sample ratio selection problem in MNMT as an optimization problem based on the Double Power Law. In our experiments, it achieves better performance than temperature searching and gradient manipulation methods with only 1/5 to 1/2 of the total training budget. We release the code at https://github.com/pkunlp-icler/ParetoMNMT for reproduction.

## 1 Introduction

Multilingual Neural Machine Translation (MNMT) enables multiple translation directions in a single model, significantly reducing deployment costs of translation systems and benefiting low-resource directions (Sennrich et al., 2015). MNMT can be framed as a multi-task learning problem, where we expect to improve the overall performance across translation directions. However, it is infeasible to find an optimal solution for all translation tasks because of the negative interference among translation directions (Lin et al., 2019; Yu et al., 2020; Wang et al., 2021). A common method to make trade-offs among different languages is scalarization, where a combination of normalized weights is assigned to the loss of each direction during training. Xin et al. (2022) empirically show that different weight combinations can result in Pareto optimal solutions, where no one solution could outperform another in all tasks. All such solutions could form a Pareto front as shown in Figure 1. Fernandes et al. (2023) show that there is a power law between the weight of one direction and its final generalization performance, indicating a positive relation between weight and performance.

In this work, we empirically find that the positive relation together with the power law from Fernandes et al. (2023) does not hold true in a more realistic MNMT scenario where the data for different directions is highly imbalanced. The performance of a direction may decrease as its weight increases.

---

[*]Equal Contribution.
[†]Corresponding author.

37th Conference on Neural Information Processing Systems (NeurIPS 2023).

It raises our interests that 1) What causes the inconsistency phenomena? 2) Can we model the trade-off behavior among different directions? 3) How should we set the weights for each direction?

To answer these questions, we first identify the problem in MNMT that the scalarization method could not generate Pareto optimal solutions when there exist low-resource directions, which is named **the collapse of Pareto front**. Secondly, to quantify how the performance of a given direction changes with its sampling ratio, we propose the Double Power Law. The law takes the number of training data into consideration and explicitly models the relationship between capacity occupation and the potential of over-fitting in a given direction, which has the form:

$$\mathcal{F}_i(p, D_i) = \underbrace{(k \cdot p)^{-\alpha}}_{\text{Capacity Occupation}} + \underbrace{(D_i^\gamma + b) \cdot (q \cdot p)^\beta}_{\text{Intrinsic Over-fitting}} + \underbrace{M_\infty^{(i)}}_{\text{Bias Term}} \tag{1}$$

where $\mathcal{F}_i$ is the predicted performance of task $i$ measured by cross-entropy loss, $p$ is the sampling ratio, $D_i$ is the number of training examples and $M_\infty^{(i)}$ is a constant bias term. The rest are fixed parameters to estimate. Experiments justify the robustness of the law under various settings.

Based on the Double Power Law (DPL), we propose an approach to compute the optimal sampling ratios. Our experiments demonstrate that DPL outperforms directly search for the optimal temperature and gradient manipulation methods for MNMT, achieving better accuracy (+0.3 BLEU) with only 1/5 to 1/2 training cost in our experiments.

To summarize, the contribution of this work is threefold:

1. We identify and analyze the collapse of Pareto Front phenomenon in MNMT.
2. We propose the Double Power Law to model the trade-off among different directions in MNMT, which is robust on languages, data adequacy, and number of directions.
3. Based on the Double Power Law, we propose a method to automatically compute the optimal sampling ratio combination, eliminating the need for manual tuning of the sampling temperature and achieving better performance in BLEU, COMET and BertScore metrics.

## 2 Collapse of Pareto Front in Multilingual Neural Machine Translation

### 2.1 Pareto Front in MNMT

**Pareto Front** In multi-task learning, where $K$ is the number of objectives and $\mathcal{L}_i(\boldsymbol{\theta})$ is the loss of task $i$ under solution parameterized by $\boldsymbol{\theta} \in \mathbb{R}^n$, we say solution $\theta_2$ **Pareto dominates** solution $\theta_1$ if $\forall\, 1 \le i \le K, \mathcal{L}_i(\theta_2) \le \mathcal{L}_i(\theta_1)$ and $\exists\, 1 \le i \le K, \mathcal{L}_i(\theta_2) < \mathcal{L}_i(\theta_1)$. The **Pareto front** is a set of solutions where no solution Pareto dominates another. As shown in Figure 1, we give two toy examples of the possible shape of Pareto front when $K = 3$ and $K = 2$.

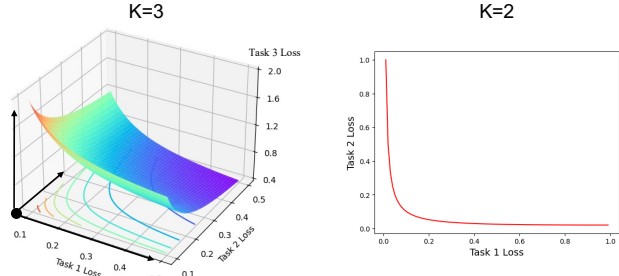

Figure 1: Toy examples of Pareto front. When there are three tasks (K=3), we color the Pareto front according to task 3's loss. We also plot the contours, which are also Pareto fronts in 2-dimension.

Scalarization (Boyd and Vandenberghe, 2004) is a traditional method to optimize the multi-task learning model where a set of predefined normalized weights $\boldsymbol{w}$ is applied to each task. *If there is little interaction among different tasks*, that one task's loss would drop when increasing its weight, equation 2 with different $\boldsymbol{w}$ could guarantee solutions that can form a Pareto front (Xin et al., 2022).

$$\hat{\boldsymbol{\theta}}(\boldsymbol{w}) = \arg\min_{\boldsymbol{\theta}} \mathcal{L}(\boldsymbol{\theta}; \boldsymbol{w}) \quad \text{s.t.} \quad \mathcal{L}(\boldsymbol{\theta}; \boldsymbol{w}) = \sum_{i=1}^K w_i \mathcal{L}_i(\boldsymbol{\theta}), \quad \boldsymbol{w} > 0, \quad \sum_i w_i = 1 \tag{2}$$

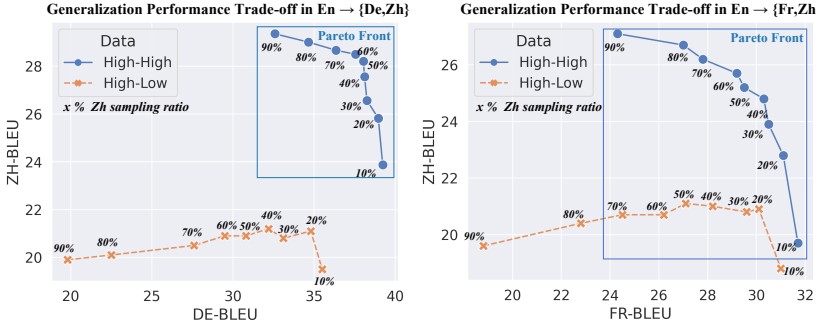

Figure 2: Generalization performance trade-off curves under different data distributions. The blue line stands for English→{German-4.6M, Chinese-10M} (left) and English→{French-10M, Chinese-10M}(right). The orange line stands for the result when replacing the high-resource English→Chinese-10M to English→Chinese-260K. The Pareto front collapses (sampling ratio increases while performance drops) when changing the high-resource direction to a low-resource one.

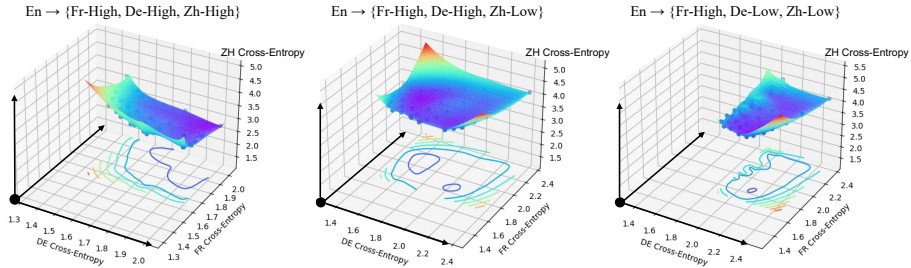

Figure 3: Generalization Performance trade-off curves under different data distributions under the 3-direction setting. The left figure shows a Pareto front where one direction's performance will always decrease when another direction's performance increases. However, the performance of low-resource direction would increase in step with the high-resource direction's at some points.

**Multi-task Learning for MNMT** MNMT can be framed as a multi-task learning problem where one model can do the translation among multiple directions (Dong et al., 2015; Aharoni et al., 2019). To prevent the high-resource directions from dominating the training and to improve the low-resource directions' performance, scalarization is implemented through proportional sampling, where the sampling ratio of examples for task $i$ in a training batch is $w_i$. Currently, there is no consensus on how to directly set specific $w_i$ for each task, temperature-based sampling is more commonly used where $w_i = \frac{p_i^{1/T}}{\sum_j^K p_j^{1/T}}$, $p_i$ denotes the ratio of the training examples from task $i$.

## 2.2 The Collapse of Pareto Front Widely Exists

We use datasets provided in WMT10 (Wang et al., 2020b) and WMT19 (Barrault et al., 2019) to conduct the MNMT experiment. The description of dataset is listed in Appendix A. In the 2-task setting, we experiment with the sampling ratio of one task ranging from 10% to 90%. In the 3-task setting, we experiment with each task ranging from 10% to 80% resulting in 32 different ratio combinations. More detail about the training is in Appendix B. The experiment results and major findings in this section are illustrated in Figure 2,3 and 4.

**The collapse exists in different language pairs** In Figure 2, we can see that the generalization performance trade-off situation is different between the High-High and High-Low data settings. Under the High-High setting, the solutions from scalarization could form a Pareto front where one task's performance increases with the decrease in the other task. This is expected and also in line with the recent results from Xin et al. (2022); Fernandes et al. (2023). However, the Pareto front disappears when we replace the English→Chinese-10M with English→Chinese-260K. Fernandes et al. (2023) restrict the directions to high-resource ones so they also get Pareto optimal results, Xin et al. (2022)

also find that there exists a global optimal point in imbalanced multilingual translation however do not give it an explanation. Understanding the reason can help build a better MNMT system since the multilingual training data is highly imbalanced in reality(Haddow et al., 2021; team et al., 2022).

In Figure 2, when increasing the sampling ratio for one High-Resource direction, its performance continually goes up. While increasing the sampling ratio for one Low-Resource direction from 10% to 90%, its performance first goes up and then gradually goes down. This is the direct cause of the collapse of Pareto front. The collapse also exists when there are more than two languages. As shown in Figure 3, we draw the 3-task trade-off front using linear interpolation. The front in the left where all three directions are high-resource, is most similar to the example 3D Pareto front in Figure 1. When there exists low-resource directions as shown in the middle and the right in Figure 3, the Pareto front would disappear.

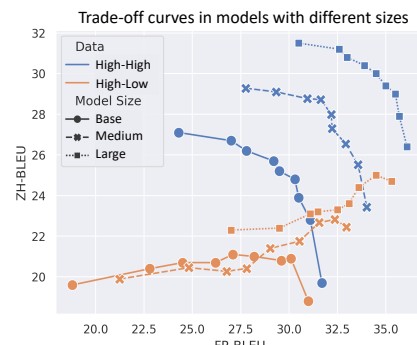

Figure 4: Performance (BLEU) trade-off curves for English→{French, Chinese} under different model sizes and data distributions.

It indicates that the relation between High and Low resources directions is not a zero-sum game and there exists a sampling ratio range where different tasks can improve or deteriorate simultaneously. For example, in Figure 2, this range could be 40% to 90% sampling ratio for the low-resource English-to-Chinese direction.

**The collapse exists in models with different sizes**    Figure 4 shows the trade-off curves on models with different sizes. The Pareto front would collapse when there exist low-resource directions.

The collapse of Pareto front complicates the optimization of MNMT because the sampling ratio must be carefully set or the performance might fall in the collapsed area where the model's capacity is not fully exploited and there exist other solutions that can Pareto dominates the current performance.

## 2.3   Explore the Reason for the Collapse

The collapse is closely related to inadequate data. If we only consider the performance of low-resource direction in Figure 2, the performance of the low-resource direction briefly improves with an increase in the sampling ratio from 10% to 20%, but then gradually deteriorates as the sampling ratio continues to increase from 20% to 90%. This is in contrast to the performance of high-resource directions, which continue to improve with an increase in sampling ratio from 10% to 90%.

The effect of increasing the sampling ratio for a task is twofold: on the one hand, it gives it more weight and the model would allocate more capacity to optimize its performance, on the other hand, it also increases its risk of over-fitting. For directions with abundant training examples, the over-fitting term is less significant so its performance would keep improving with more sampling ratio. However, for low-resource directions, the risk of over-fitting would be dominant for a high sampling ratio because of the lack of training examples. A small sampling ratio serves as a form of regularization for these low-resource

| Ratio(De:Hi) | Encoder Last | Decoder Last |
|---|---|---|
| 1:9 | 21.80/30.96 | 32.96/42.98 |
| 5:5 | 11.87/20.12 | 25.88/33.59 |
| 9:1 | **10.43/14.48** | **23.88/27.86** |

Table 1: The sharpness for En→{De} / En→{Hi} tasks. The result indicates that with more high-resource data, the model tends to come to a flatter local minima after training.

directions. This difference accounts for the contrasting performance trends observed between low-resource and high-resource directions.

It has been shown that the model would be less likely to over-fit and generalize better when it reaches a flatter local minimum after the gradient descent optimization (Keskar et al., 2016; Kleinberg et al., 2018; Foret et al., 2020). To justify our assumption, following Keskar et al. (2016), we compute the partial local curvature i.e. sharpness of the optimized model through the trace of the Hessian matrix as shown in Equation 3, where $X$ stands for held-out evaluation data and $Y$ stands for the label. We compute the sharpness of the FFN layer at different transformer layers.

$$Sharpness(f_\theta, X, Y) = tr(\nabla^2 f_\theta(X, Y)) \tag{3}$$

As shown in Table 1, we compute the sharpness of our MNMT model on English→{German, Hindi} task with different sampling ratio combinations. The result shows that with more sampling ratio for high-resource direction, the model could reach a flatter local minimum for both tasks.

The results of the generalization performance and sharpness experiments indicate that adding weight for a certain translation direction is a double-edged sword. It forces the model to optimize more towards minimizing the training loss of this task, at the same time increasing its over-fitting risk, especially for the low-resource directions.

## 3 The Double Power Law

In the previous section, we discussed the collapse of Pareto front phenomena and its impact on MNMT. The collapse presents a challenge for determining the sampling ratio combination, as a poor combination can adversely affect the final performance by deviating from potential Pareto solutions. Additionally, we identified over-fitting as a significant problem in MNMT training especially for low-resource directions. While a straightforward solution would be to grid search the sampling ratio combination, it becomes impractical with increasing directions due to the exponentially growing search space. In this section, we are going to answer the following questions:

***Is there a pattern behind the collapse phenomena? Can we predict the performance trade-off curve among different directions?***

Fernandes et al. (2023) adopt single power law inspired by the scaling literature in NMT (Ghorbani et al., 2021b; Gordon et al., 2021) to predict how the sampling ratio affect the performance of MNMT. However, it assumes that all directions have unlimited data, which does not reflect the diverse data adequacy in reality for supervised MNMT training. Under Fernandes et al. (2023)'s power law, the trade-off curve of any two directions can form a Pareto front, which could not capture the collapse.

We find out that: the generalization performance $\mathcal{F}$ of the $i$-th direction in MNMT follows a double power law, which takes the sampling ratio $p$ and the number of training examples $D_i$ as input:

$$\mathcal{F}_i(p, D_i) = \underbrace{(k \cdot p)^{-\alpha} + L_\infty^{(i)}}_{\text{Capacity Occupation}} + \underbrace{\mathcal{G}^\downarrow(D_i) \cdot (q \cdot p)^\beta + O_\infty^{(i)}}_{\text{Intrinsic Over-fitting}} \tag{4}$$

where $\mathcal{F}_i$ is measured through evaluation cross-entropy loss. $k, \alpha, q, \beta$ are fixed parameters, $\mathcal{G}^\downarrow(D)$ is a monotonically decreasing function of the number of training examples $D_i$. $L_\infty^{(i)}$ and $O_\infty^{(i)}$ are bias terms that are relevant to the direction and the model.

### 3.1 Capacity Occupation and Intrinsic Over-fitting Terms

Based on our previous discussion, the generalization performance for a certain direction not only depends on its sampling ratio but also on its number of training examples. So we set two terms to jointly model the generalization performance, namely Capacity Occupation term and Intrinsic Over-fitting term. As shown in Equation 4, similar to the power law in Fernandes et al. (2023), **Capacity Occupation term** can reflect that with a larger sampling ratio, the model would assign more capacity for the task to minimize its training loss, which could possibly improve the generalization performance. **Intrinsic Over-fitting term** reflects the fact that with a larger sampling ratio, the model is more likely to over-fit on this task, which would harm its generalization performance. Meanwhile, the scale of Intrinsic Over-fitting term is negatively related to the number of training examples. With fewer training examples, the influence of Intrinsic Over-fitting would be more significant.

### 3.2 Parameter Estimation

We estimate the parameters of the double power law in three steps. We conduct a series of English→ {French-10M, German-260K} experiments with English→French-10M sampling ratio ranging from 10% to 90%. 1) We estimate the parameters of the Capacity Occupation term with English→French's results by ignoring the Intrinsic Over-fitting term because it has minor effect when $D_i$ is large for high-resource direction. 2) We fix the parameters of the Capacity Occupation term and estimate the parameters of the Intrinsic Over-fitting term with the results of

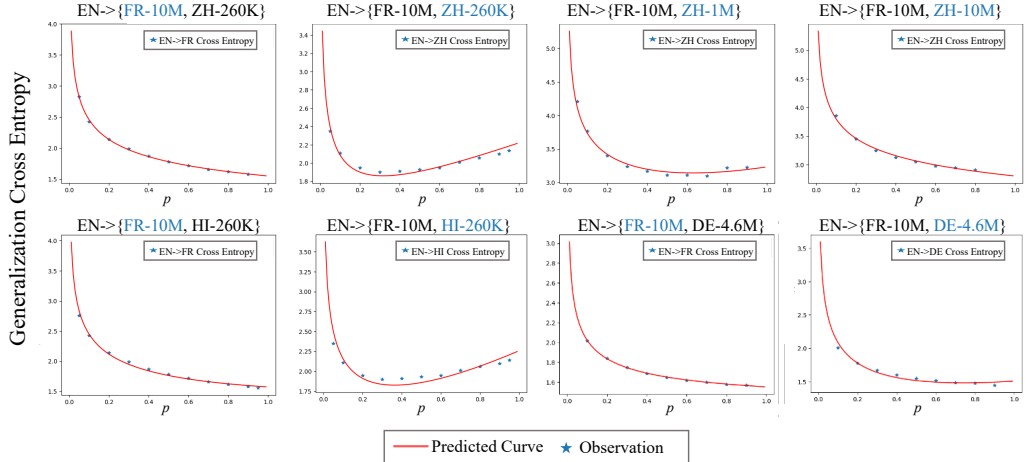

Figure 5: Evaluation of Equation 6 under different languages and data-adequacy settings. The curve predicts how one direction's generalization performance changes with its sampling ratio. The Double Power Law is robust under different scenarios.

English→German-260K. 3) We fix all parameters in Equation 4 except for the value of $\mathcal{G}^{\downarrow}(D)$. We suppose that $\mathcal{G}^{\downarrow}(D)$ also subjects to the form of power law and conduct the same series of experiments on English→ {French-10M, German-1M} and English→ {French-10M, German-4.6M} to get values of $\mathcal{G}^{\downarrow}(D)$ given different $D_i$ and estimate the parameters in $\mathcal{G}^{\downarrow}(D)$. Thus, the double power law could be summarized below, where $M_\infty^{(i)}$ stands for the sum of bias terms $L_\infty^{(i)}$ and $O_\infty^{(i)}$:

$$\mathcal{F}_i(p, D_i) = \underbrace{(k \cdot p)^{-\alpha}}_{\text{Capacity Occupation}} + \underbrace{(D_i^\gamma + b) \cdot (q \cdot p)^\beta}_{\text{Intrinsic Over-fitting}} + \underbrace{M_\infty^{(i)}}_{\text{Bias Term}} \tag{5}$$

Following the three steps, we estimate[1] the parameters of the double power law for our base-size model. For convenience, we present the estimated double power law below:

$$\mathcal{F}_i(p, D) = (0.07 \cdot p)^{-0.20} + \left(D^{-0.33} - 0.50\right) \cdot (1.18 \cdot p)^{1.21} + M_\infty^{(i)} \tag{6}$$

The law indicates an interesting result: the generalization performance trade-off among directions in MNMT is mostly relative to the training data size in each direction. The difference in languages only influences the bias term.

### 3.3 Evaluation and Discussion

We evaluate the estimated double power law in different task and data size combinations as shown in Figure 5. For different task settings, we only tune the bias term $M_\infty^{(i)}$ in Equation 6. As shown in Figure 5, the double power law can generally well reflect how the performance of one task changes with its sampling ratio for both of the high-resource and low-resource directions.

**Generalization to more directions**  We also evaluate the double power law when there are more than 2 languages. We conduct a three-direction MNMT experiment on English→ {French-10M, German-4.6M, Chinese-260K}, where English→ Chinese is the low-resource direction. We fix the sampling ratio of English→ German to 0.1 and change the sampling ratio of English→ French from 0.1 to 0.8 and tune the bias term $M_\infty^{(i)}$ in Equation 6 to fit the performance curve of English→ French and English→ Chinese. As shown in Figure 6, the double power law can also well reflect the performance trade-off with more directions.

---

[1]we use scipy.optimize.curve_fit function from the scipy library. The unit of training examples is million.

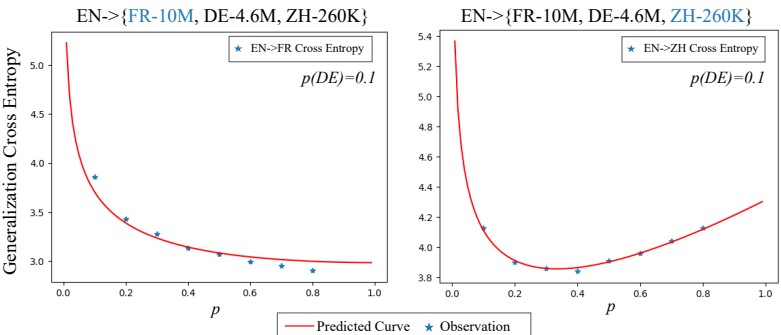

Figure 6: We examine Equation 6 under the 3-direction setting. Double Power Law can well describe the performance trend of both the high-resource (FR-10M) and low-resource directions (ZH-260K).

**The trade-off curve is language-agnostic**     As shown in Figure 5and 6, by adjusting the bias term in Equation 6, the predicted curve can well describe the trade-off with different languages setting, both the similar(FR and DE) and the distant (FR and ZH) ones. In other words, the shape of the trade-off curve is more related to the number of training examples in each direction rather than the language of the direction, which mostly influences the bias term of the curve. This result is in line with Xin et al. (2022); Shaham et al. (2022) that the training data size plays a more important role than language similarity in the trade-off of different directions' performance.

**Critical point for low-resource directions**     When we look at the predicted performance curve and the observations for low-resource directions, it all shows a "U" shape. It means there exists a critical point when increasing the sampling ratio for low-resource direction. On the right side of the critical point, the Capacity Occupation term has more influence on the generalization performance. The Intrinsic Over-fitting term becomes dominant after the sampling ratio goes beyond the critical point. While for high-resource directions, the curve is monotonic.

### 3.4   Application of the Double Power Law

We adopt the Double Power Law to compute the optimal sampling ratio combination for MNMT. For a task with $n$ directions, it is evaluated by the weighted sum of each direction's performance. $r_i$ is the weight of the $i$-th task and $\sum_{i=1}^{n} r_i = 1$.

Given the evaluation metrics' weights $r$ and the number of training examples in each task $d$, the goal is to compute the optimal sampling ratio combination $\hat{p}$, where we assume that all directions follow the double power law $\mathcal{F}$ in Equation 4. Thus, the optimization objective becomes:

$$\hat{p} = \arg \min_{p} \mathcal{L}(p; r; d) \quad \text{s.t.} \quad \mathcal{L}(p; r; d) = \sum_{i=1}^{n} r_i \mathcal{F}_i(p_i, d_i), \quad p > 0, \quad \sum_{i} p_i = 1 \quad (7)$$

## 4   Experiments

### 4.1   Settings

We conduct experiments on a two-direction English $\rightarrow$ {German-4.6M, Hindi-260K} task and a four-direction English $\rightarrow$ {French-10M, German-4.6M, Chinese-260K, Hindi-260K} task using the same training configuration as in Section 2.2 in our base size model.

We use two different evaluation metrics: 1) Arithmetic mean ($r_i = \frac{1}{n}$). 2) Customized metric where we are interested in improving the performance of low-resource direction $i$ ($r_i = 1, r_{j \neq i} = 0$).

We solve Equation 7 with the estimated double power law as in Equation 6. Even though we do not know the bias term $M_{\infty}$ in Equation 6, it does not influence the computation of the optimal sampling ratio combination $\hat{p}$. With the computed $\hat{p}$, we train the MNMT model according to Appendix B.

(a) 4-direction (Arithmetic mean)

| Methods | Cross-Entropy ↓ | | | | | BLEU ↑ | | | | | TrainingTime |
|---|---|---|---|---|---|---|---|---|---|---|---|
| | FR | DE | ZH | HI | AVG | FR | DE | ZH | HI | AVG | |
| *Temperature-based Sampling* | | | | | | | | | | | |
| T=1 | 1.58 | **1.55** | 4.73 | 3.21 | 2.77 | **28.3** | 28.2 | 15.7 | 6.4 | 19.7 | 1.0x |
| T=2 | 1.59 | 1.61 | 3.93 | 2.59 | 2.43 | 27.7 | 28.1 | 20.5 | 10.9 | 21.8 | 1.0x |
| T=5 | 1.65 | 1.63 | 3.83 | 2.56 | 2.42 | 27.2 | 28.0 | 21.1 | 10.8 | 21.8 | 1.0x |
| T=10 | 1.70 | 1.76 | **3.77** | **2.48** | 2.43 | 25.8 | 27.2 | 21.8 | **11.4** | 21.6 | 1.0x |
| T=100 | 1.75 | 1.83 | 3.79 | 2.54 | 2.48 | 25.4 | 26.7 | **22.1** | 11.0 | 21.3 | 1.0x |
| *Gradient Manipulation* | | | | | | | | | | | |
| PCGrad | **1.57** | 1.59 | 4.78 | 3.28 | 2.81 | 28.1 | **28.5** | 15.3 | 6.3 | 19.6 | 3.2x |
| Gradient Vaccine | 1.58 | 1.62 | 4.94 | 3.41 | 2.89 | 28.0 | **28.5** | 15.0 | 5.9 | 19.4 | 3.2x |
| DPL (Arithmetic mean) | 1.64 | 1.63 | 3.83 | 2.53 | **2.41** | 27.3 | 28.0 | 20.9 | 11.3 | **21.9** | 1.0x |

(b) 2-direction (Arithmetic mean & customized metric)

| Methods | Cross-Entropy ↓ | | | BLEU ↑ | | | Training Time |
|---|---|---|---|---|---|---|---|
| | DE | HI | AVG | DE | HI | AVG | |
| *Temperature-based Sampling* | | | | | | | |
| T=1 | **1.41** | 3.20 | 2.31 | **36.6** | 8.4 | 22.5 | 1.0x |
| T=2 | 1.44 | 2.56 | 2.00 | 35.8 | 11.1 | 23.4 | 1.0x |
| T=5 | 1.55 | 2.51 | 2.03 | 34.1 | 12.2 | 23.1 | 1.0x |
| T=10 | 1.60 | 2.59 | 2.10 | 33.0 | 11.7 | 22.3 | 1.0x |
| T=100 | 1.68 | 2.69 | 2.19 | 27.1 | 10.0 | 18.5 | 1.0x |
| *Gradient Manipulation* | | | | | | | |
| PCGrad | 1.42 | 3.18 | 2.30 | **36.6** | 8.3 | 22.5 | 2.1x |
| Gradient Vaccine | 1.43 | 2.95 | 2.19 | 36.4 | 9.8 | 23.1 | 2.1x |
| DPL (Arithmetic mean) | 1.46 | 2.52 | **1.99** | 35.4 | 12.0 | **23.7** | 1.0x |
| DPL ($r_{HI} = 1$) | 1.57 | **2.45** | 2.01 | 34.4 | **12.6** | 23.5 | 1.0x |

Table 2: Results of the main experiments for the application of Double Power Law. We report the average results of three random seeds for each method.

## 4.2 Baselines

**Temperature-based Sampling** We search $T \in \{1, 2, 5, 10, 100\}$, which are most frequently used in NMT literature. After choosing T, we set the sampling ratio $p'_i = p_i^{1/T} / \sum_j^K p_j^{1/T}$ where $p_i$ denotes the ratio of the training examples from task i to the whole training set.

**Gradient Manipulation** We also compare our method to gradient manipulation methods that are reported to improve MNMT performance: 1) PCGrad(Yu et al., 2020) and 2) Gradient Vaccine(Wang et al., 2021). We employ the default configuration of the methods described in their corresponding papers. Gradient manipulation methods require computing the cosine similarity of gradients at each optimization step and adjusting the gradient direction to resolve conflicts, resulting in an increased computational budget along with the increase of directions.

## 4.3 Results

As shown in Table 2, we report both the generalization cross-entropy and detokenized BLEU score using sacrebleu[2] for different sampling methods.

Under the arithmetic mean metric, DPL reaches similar generalization cross-entropy compared to the best result when tuning temperature (T=2) and gains +0.3 average BLEU scores than T=2 in the 2-direction setting. In the 4-direction setting, DPL also gains better results(+0.1 BLEU) compared to the best result when tuning temperature (T=5). The best sampling temperature varies for different MNMT tasks while DPL could surpass the corresponding best results by just computing once, reducing the searching time to 1/5 of origin.

Comparing to the Gradient Manipulation methods, DPL leads to a much better average performance than the best method in both experiments (+0.6 BLEU in 2-direction , +2.3 BLEU in 4-direction),

---

[2] nrefs:1|case:mixed|eff:no|tok:13a|smooth:exp|version:2.1.0

| Metric | Method | FR | DE | ZH | HI | AVG |
|--------|--------|----|----|----|----|-----|
| BERT-Score | T=5 (best T) | **71.2** | **74.4** | 70.5 | 65.5 | 70.4 |
|            | DPL(Ours) | 71 | 74.1 | **71.6** | **66.3** | **70.8** |
| COMET | T=5 (best T) | **54.4** | **52.4** | 46.8 | 45.2 | 49.7 |
|       | DPL(Ours) | 53.9 | 52.1 | **48.2** | **46.5** | **50.2** |

Table 3: Model-Based evaluation results of the 4-direction multilingual NMT experiments. DPL method consistently outperforms the best temperature-based method in terms of average results.

with only 1/2 and 1/3 training time. The result is also in line with Xin et al. (2022) that simple scalarization could beat the complex optimization method if we set the weights properly.

We also evaluate the customized metric in the 2-direction experiment, where we hope to best enhance the performance of the low-resource direction. It turns out that DPL is more effective (+0.4 BLEU) in improving the low-resource direction than tuning the temperature (p<0.05).

In addition to using the BLEU score for evaluation, we also utilize model-based evaluation techniques, specifically Bert-Score (Zhang et al., 2020) and COMET (Rei et al., 2020). Research indicates that these metrics often align more closely with human evaluations than their model-free counterparts. As detailed in Table 3, DPL outperforms the best temperature-based method, showing an average improvement of 0.4 in BERT-Score (p<0.05) and 0.5 in COMET score (p<0.05).

The results justify that through the Double Power Law, we can compute the sampling ratio combination that best satisfies our expectation *without manually searching the sampling ratios.*

### 4.4  Discussion

**The Double Power Law of model with different sizes**    As shown in Table 4, we evaluate the how the power terms of Double Power Law change in models with different sizes.

Among the parameters in the Double Power Law, we are more interested in power terms in the Capacity Occupation($\alpha$) and Intrinsic Over-fitting($\beta$) term since they better reflect the sensitivity of performance changes against sampling ratio. We keep the other parameters the same as in Equation 6 and estimate the power terms in the Medium and Large size models.

| Model Size | $\alpha$ | $\beta$ |
|------------|----------|---------|
| Base(64M) | 0.20 | 1.21 |
| Medium(102M) | 0.25 | 2.41 |
| Large(317M) | 0.27 | 3.43 |

Table 4: The estimated power terms of for different models.

It shows that the power term($\beta$) in the Intrinsic Over-fitting term increases rapidly with the growth in model size. While the power term($\alpha$) in the Capacity Occupation term changes much slower. For larger model, over-fitting phenomena of low-resource direction will be more severe and the final performance would be more likely to fall into the collapsed area when increasing the sampling ratio for low-resource directions.

**Generalization of DPL to more diverse Directions**    To better test the generality of DPL computed as Equation 6, we ran experiments on 10 languages in X-EN from WMT10 (Callison-Burch et al., 2010). Dataset details are in Appendix C. Figure 7 indicates a Pareto Front Collapse in low-resource directions, with the double power law fitting well across data/language conditions ($r^2$ values of 0.86 and 0.92 for low and high resources). As for the difference, we observed a more pronounced pareto front collapse in X-EN than in EN-X. By only adjusting the $\gamma$ term in DPL to -0.25, the fit improves for X-EN directions ($r^2$: 0.86 to 0.91 for low-resources; 0.92 to 0.97 for high-resources), highlighting the significant effect of the Intrinsic Over-fitting term in X-EN directions.

The results combined show that we can rely on the instantiated DPL when the model and the training hyper-parameters (e.g., batch size, total update steps ) are not changed. The DPL is robust when introducing more languages or changing the translation directions.

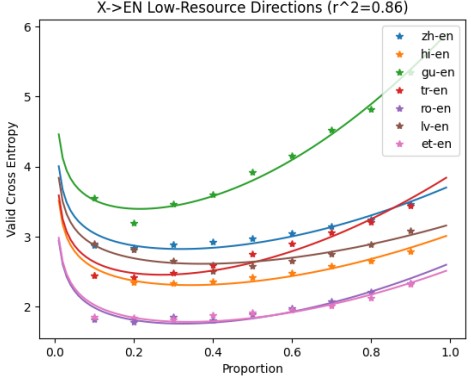 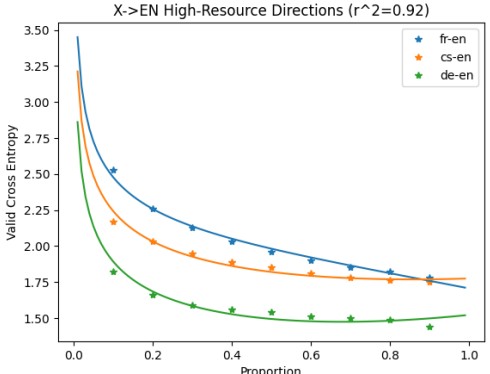

(a) As the proportion increases, the validation loss goes down then goes up for low-resource languages.

(b) The validation loss goes down when the proportion of high-resource direction increases.

Figure 7: Evaluation of the Double Power Law under X->EN translation setting, with more diverse languages. The points are the actual value and the solid lines are the predicted values for one direction at certain sampling ratio.

## 5 Related Work

Many previous work studied the trade-off among different directions and how to achieve a better overall performance in MNMT. Aharoni et al. (2019); Shaham et al. (2022) empirically study how different settings influence the positive and negative interference among different directions. Xin et al. (2022)firstly incorporate Pareto optimity to the analysis of MNMT system. Fernandes et al. (2023) propose to use a single power law to model how one direction's performance changes with its sampling ratio under the assumption that all directions have unlimited data.

Another related line of works study multi-task optimization for MNMT. Yu et al. (2020); Wang et al. (2021) propose to dynamically altering the conflict gradients eliminate the negative interference among tasks. Wang et al. (2020a); Zhou et al. (2021) explore how to dynamically adjust the optimization objective along with the training process to better schedule the training for different directions. Li and Gong (2021) focus on the imbalance nature of MNMT task and propose an optimization method based on the local curvature of the loss landscape. However, Xin et al. (2022) recently show that simple scalariaztion method could beat many multitask optimization methods for MNMT if the sampling ratio for different tasks are correctly set.

While we focus on how to set the optimal weights, modeling how the size of neural network influences the performance also attracts much attention recently, i.e. the neural scaling law (Kaplan et al., 2020; Hoffmann et al., 2022; OpenAI, 2023). In the field of NMT, Ghorbani et al. (2021a); Gordon et al. (2021) show that the scaling behavior of cross-entropy loss follows a power law with the model size as input while Fernandes et al. (2023) extends the conclusions to multilingual setting.

## 6 Conclusion

In this work, we start from our observations that the trade-off front is no longer Pareto front in Multilingual Neural Machine Translation when there exists low-resource directions, i.e. the collapse of Pareto front. We analyse the reason causing the phenomena and find that there exists a balance between model capacity occupation and the intrinsic risk to over-fit for certain direction. Based on the findings, we propose the Double Power Law to model how the performance of a given direction changes with its sampling ratio, which is robust across different task settings and is capable of capturing the collapse phenomena. Finally, we frame the sampling ratio selection problem for Multilingual Neural Machine Translation as an optimization problem, which greatly reduces the cost for manually tuning the sampling temperature and reaches better results.

# 7 Acknowledgements

We thank all reviewers for their helpful advice. This paper is supported by the National Key Research and Development Program of China under Grant No.2020AAA0106700, the National Science Foundation of China under Grant No.61936012.

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

## A    Datasets for Main Experiments

| Direction | Dataset | Num. of Training Examples | Evaluation Dataset |
|---|---|---|---|
| English-to-French | WMT10 | {10M} | newtest15 |
| English-to-Chinese | WMT19 | {10M, 1M, 260K} | newtest19 |
| English-to-German | WMT10 | {4.6M, 1M, 260K} | newtest18 |
| English-to-Hindi | WMT10 | {260k} | newtest14 |

Table 5: The datasets description for the main experiments. We randomly choose a subset of the full training set of a direction to form a smaller one.

## B    Training Details of Multilingual NMT experiments

In this section, we would provide concrete details of our experimental setup for reproduction.

Similar to Xin et al. (2022)'s setting, we conduct our experiments on three Transformer (Vaswani et al., 2017) models with different sizes as shown in Table 6

| Size | Enc. Layer | Dec. Layer | Hid. Dim | FFN Dim | Max Steps | Max Tokens per Batch |
|---|---|---|---|---|---|---|
| Base (64M) | 3 | 3 | 512 | 2048 | 100k | 20k |
| Medium (102M) | 6 | 6 | 512 | 2048 | 100k | 20k |
| Large (317M) | 6 | 6 | 1024 | 4096 | 400k | 80k |

Table 6: Overview of model sizes and optimization hyper-parameters. The Large-size model is trained with a larger batch-size and training steps or it would fail to converge. For 4-task experiment, we double the max training steps due to the increase of training data.

For all of our model, we use a 64k sentence piece vocabulary. All models are trained[3] with 4k warm-up steps with the learning rate linearly increasing from $0$ to $3e^{-4}$ then decreasing with inverse_sqrt learning rate scheduler. The label smoothing term is set to 0.1 following the NMT literature convention. Evaluation is done every 5k steps, and we choose the best checkpoint with lowest average validation loss before examining the final generalization performance.

## C    Datasets for Generality Testing

| Language Pair | Num. of Training Examples | Evaluation Dataset |
|---|---|---|
| fr-en | 10M | newtest13 |
| cs-en | 5M | newtest16 |
| de-en | 4.6M | newtest16 |
| zh-en | 260K | newtest19 |
| hi-en | 260K | newtest14 |
| gu-en | 80k | newtest19 |
| tr-en | 180k | newtest16 |
| ro-en | 500K | newtest16 |
| lv-en | 300K | newtest17 |
| et-en | 300k | newtest18 |

Table 7: Dataset description for multilingual x-en translation experiments. The dataset is taken from WMT10. The dataset includes languages from diverse language family and data-adequacy.

---

[3]We use fairseq(Ott et al., 2019) as the training framework

