# OpenReview forum: "On the Pareto Front of Multilingual Neural Machine Translation"
_NeurIPS.cc/2023/Conference — NeurIPS 2023 poster_

### Official Review · Reviewer_CMx8 · 2023-06-14

**Soundness:** 3 good
**Presentation:** 3 good
**Contribution:** 2 fair
**Rating:** 6
**Confidence:** 4

**Summary:**

This paper frames the sampling ratio selection problem of MNMT as an optimization problem, and introduces a double power law to control the sampling ratio for different languages. The author argue that traditional method assumes unlimited training data which does not hold true for low resource settings, thus both model capacity occupation and intrinsic risk of overfitting should be taken into consideration during training data sampling.
Experiments on multiple language direction shows better accuracy (+0.3 BLEU) with only half training cost.


**Strengths:**

This paper is an upgrade on the Fernandes et.al 2023 paper, from single power law to double power law,
taking both model capacity occupation and intrinsic risk of overfitting into consideration.
I think over the years the sampling strategy of MNMT has always been on data balancing, while the overfitting risk for low resource language has largely been ignored. And this paper addresses an important problem in this direction.
I like the organization of this paper, starting from an empirical phenomena, find the problem and solve it with a well-grounded method.


**Weaknesses:**

However, I believe there is still room for improvement in this paper:
1.  I think the double power law is still not enough to resolve the sampling problem. One important aspect is the data quality: so in the low resource setting, the quality of training data is often worse comparing with the rich resource setting, and the noisy data would also cause the performance drop (or local optimum ) during training. So if we are talking about a more realistic MNMT scenario, the data quality issue should definitely be considered.  I would like to see more analysis in this direction.

2. There are some confusing claims in the paper :  A. in the intro: "achieving better accuracy (+0.3 BLEU) with only half training cost".
But in the table 2, the statics shows something different : for example in (a),  comparing with temperature based method,  BLEU +0.1, training time both 1.0x,  with Gradient based method, BLEU +2.4, training time 1.0x vs 3.2x.   So I am not sure which one is the key comparison to look at.  Besides, when comparing the training cost, the time to estimate the parameters of the double power law should also be included.  B. in abstract, the author claims "by training over 200 multilingual models", but there is no evidence in the experiments, only a two-direction model and a four-direction model.   So I think the author should be more careful in making these claims.

3. The improvement over traditional method is marginal. Comparing with Temperature based method, the proposed method doesn't show enough improvement in both BLEU score and training time.








**Questions:**

1. So if we change the corpus size of certain language direction, should the parameters of the double power be re-estimated, and what is the time cost for this?

**Limitations:**

I think the two major limitation of this paper are the ignorance of data quality issue in MNMT training and the less convincing performance comparing with temperature based methods.

---

> ### Author Rebuttal · Authors · 2023-08-09
>
> Dear reviewer, we really appreciate your effort in reviewing our paper, and thank you for your helpful comments. We are glad that you recognize our work 1) addresses an important problem in this direction 2) with good organization of the paper 3) solves the problem with a well-grounded method. We address your questions below:
>
> > W1:  I think the double power law is still not enough to resolve the sampling problem. One important aspect is the data quality: so, in the low resource setting, the quality of training data is often worse compared with the rich resource setting, and the noisy data would also cause the performance drop (or local optimum) during training. So, if we are talking about a more realistic MNMT scenario, the data quality issue should definitely be considered. I would like to see more analysis in this direction.
> >
>
> **We found that the DPL is robust across directions with different qualities.** In the original paper, to ensure the robustness of our findings, we used two kinds of low-resources data. The one like en-hi is the true low-resource direction, and the one en-zh260k is sampled from the larger en-zh10M, which is actually a high-resources direction. The data creation process was listed in Appendix A, we would make it more clear in the paper. As shown in Figure-5 from the paper, DPL can well fit the en-hi, en-zh260k, and en-zh10M data settings.  It shows that the DPL is robust under different data-quality settings and data size plays a more important role when changing the sampling ratios.
>
> To further test DPL’s robustness under different data-quality settings, we have conducted additional experiments covering 10 diverse language languages: French (FR), German (DE), Estonian (ET), Finnish (FI), Romanian (RO), Turkish (TR), Gujarati (GU), Hindi (HI), Chinese (ZH), and Latvian (LV). For X-EN directions, as shown in Figures 1 and 2 in the **general response PDF**, there is a clear Pareto Front Collapse phenomenon for all low-resource directions. And the double power law can fit well (r^2=0.86 for low-resources, r^2=0.92 for high-resources, r^2 closer to 1 suggests a better fit) under different data-adequacy/language conditions. The results confirm that the Pareto Front Collapse phenomenon and the double power law hold for both X-to-en and en-to-X directions across these diverse languages, further strengthening our claim of the power law's effectiveness in multilingual machine translation with more diverse languages, directions, and data-quality settings.
>
> > W2: There are some confusing claims in the paper.
> >
>
> Thanks for pointing out. A) When using the Temperature-based Sampling method, we need to **search** the best sampling ratio every time we have a new MNMT setting, in our experiment we search 5 different temperatures so the total training time for the Temperature-based Sampling method is 5x. While for the DPL method, the parameters of DPL **do not need to be re-estimated** in our experiment, the same DPL parameter is transferred among different experiment settings and we only need to train the model once. We would make the comparison more clear in the revised paper. B) by saying “200 multilingual models” in the abstract, we mean the total number of models trained in this paper to get all the conclusions including the Pareto Front Collapse phenomenon and the DPL. We would make it more clear in the revised paper.
>
> > W3: The improvement over the traditional method is marginal. Compared with Temperature based method, the proposed method doesn't show enough improvement in both the BLEU score and training time.
> >
>
> Compared with Temperature based method, which needs searching the best Temperature by training the models for 5 times in different MNMT experiment settings, DPL does not need re-estimate the parameters and only needs to train the model once when involving new directions or changing corpus size. In our experiments, the same DPL parameters are transferred among the 2-direction and 4-direction experiments, which is the same as Equation-6. We added an additional model-based metric BERTScore to evaluate the performance, and DPL shows consistent improvement over 0.4 points (4-direction avg 70.4 → 70.8, p<0.05) and 1.0 points (2-direction avg 75.4 → 76.4, p<0.05).  Please refer to the general response for the detailed BERTScore results.
>
> > Q1: So if we change the corpus size of a certain language direction, should the parameters of the double power be re-estimated, and what is the time cost for this?
> >
>
> When changing the corpus size of a certain direction, the parameters of DPL **do not need to be re-estimated.**  DPL already takes the corpus size as its one input variable when predicting the performance as shown in Equation 5 from the paper. The $D_i$ term is the corpus size of a certain direction.

---

> > ### Comment · Reviewer_CMx8 · 2023-08-16
> >
> > Thanks for clarifying Q1 and W1, it address some of my concerns.
> > And it's good to see some of the confusing claims mentioned in W2 will be addressed in the revised version.
> > So I decide to raise my score according to the author response.

---

> > > ### Author Response · Authors · 2023-08-18
> > > **Thanks for your helpful comments!**
> > >
> > > Thank you so much for your great efforts and time!
> > >
> > > It is a great encouragement to our work, and we are glad to see that we have addressed your concerns. And many thanks for your feedback to make our paper stronger and more solid!

---

### Official Review · Reviewer_hKkr · 2023-06-19

**Soundness:** 3 good
**Presentation:** 3 good
**Contribution:** 2 fair
**Rating:** 5
**Confidence:** 4

**Summary:**

This paper explores the influence of language pair sampling weights on the ultimate performance of Multilingual Neural Machine Translation (MNMT), specifically through the lens of the Pareto front concept. Existing literature suggests that under certain conditions, diverse sampling weights can yield solutions that constitute a Pareto front. Here, the performance of a specific task is positively related to its assigned weight.

However, this work introduces the concept of Pareto front collapse, where the performance of a low-resource direction might deteriorate as its weight escalates, especially under data-imbalance scenarios. To explain the reason for the collapse, the authors argue that increasing the sampling weight for a low-resource task not only heightens model capacity for it but also amplifies the risk of overfitting.

To predict the performance trade-off curve among different directions, the authors propose a Double Power Law (DPL) that considers capacity occupation and intrinsic over-fitting. Experimental results show that DPL captures the performnce curve successfully under various setting (including Pareto front collapse) and can used to determined the optimal sampling ratio for MNMT.

**Strengths:**

* This paper delves into a more realistic scenario of MNMT compared to previous works, wherein the data sizes of different language pairs are extremely imbalanced , and highlights the issue of the Pareto front collapse in this situation.
* The paper provides an insightful explanation for the collapse: low-resource tasks bring about the risk of overfitting as their weights are increased. This interpretation aligns intuitively with empirical results.
* The proposed DPL can predict the task performance given the data size and sampling weight which can be further used to select the optimal samlping ratios.

**Weaknesses:**

* The paper's assessment of translation performance primarily leans on BLEU, which has shown to be less reliable in terms of aligning with human judgment compared to metrics like COMET and BLEURT. This reliance on BLEU could be justified if experiments were conducted on truly low-resource languages unsupported by these advanced metrics. However, the languages featured in the study (English, German, French) are all high-resource, making this justification less feasible.
* Despite considering data imbalance scenarios, the study mostly focuses on translations from English to other languages, thus straying from a truly realistic scenario. The paper lacks supporting evidence that its findings can be generalized to more diverse cases, such as English to other languages, or non-English-centric MNMT.
* The authors have not included two crucial summaries about the DPL:
  * When can one rely on an already instantiated DPL or, conversely, when do the parameters of the DPL need to be re-estimated: e.g., when the model size changes, more language pairs are introduced, or when transitioning from English-centric to non-English-centric?
  * How many times the model needs to be retrained to re-estimate the parameters of DPL in different cases? What if I only need to estimate the sample ratios (no bias terms)?
* The training time recorded in Table 2 does not factor in time spent on hyperparameter searching, such as temperature-based sampling or DPL parameter estimation. Therefore, clarification on the above two points is crucial.

**Questions:**

Main questions are listed in *Weaknesses*. Another question here is why Figure 2 and Figure 3 describe Pareto front collapse using different indicators, BLEU, and cross-entropy.

**Limitations:**

Main limitations are listed *Weaknesses*.

---

> ### Author Rebuttal · Authors · 2023-08-09
>
> Dear reviewer, we really appreciate your effort in reviewing our paper, and thank you for your helpful comments. We are glad that you recognize our work 1) delves into a more realistic scenario of MNMT 2) provides an insightful explanation for the collapse 3) The DPL is effective in predicting the task performance. We address your questions below:
>
> > W1: The paper's assessment of translation performance primarily leans on BLEU, which has shown to be less reliable in terms of aligning with human judgment compared to metrics like COMET and BLEURT. This reliance on BLEU could be justified if experiments were conducted on truly low-resource languages unsupported by these advanced metrics. However, the languages featured in the study (English, German, French) are all high-resource, making this justification less feasible.
> >
>
> We acknowledge that BLEU scores may not provide a holistic insight into translation quality. In response to your recommendation, we have integrated the model-based metric, BERTScore (**https://github.com/Tiiiger/bert_score**), into our evaluation process. The outcomes reveal that the Pareto Front Collapse phenomenon and the BERTScore trends align with the patterns observed for BLEU scores and cross-entropy loss.
>
> In the experimental section (Table 2 of the original paper), the DPL method results in a BERTScore improvement of 0.4 points (4-direction average of 70.4 → 70.8, p<0.05) and 1.0 points (2-direction average of 75.4 → 76.4, p<0.05) when compared to the best Temperature-based Sampling method. We will incorporate these findings into the revised manuscript.
>
>  Please refer to the general response for the detailed BERTScore results.
>
> > W2: Despite considering data imbalance scenarios, the study mostly focuses on translations from English to other languages, thus straying from a truly realistic scenario. The paper lacks supporting evidence that its findings can be generalized to more diverse cases, such as English to other languages (We suppose other to English here?), or non-English-centric MNMT.
> >
>
> We have included X-EN directions and more languages such as French (FR), German (DE), Estonian (ET), Finnish (FI), Romanian (RO), Turkish (TR), Gujarati (GU), Hindi (HI), Chinese (ZH), and Latvian (LV) to the experiments.
>
> For X-EN directions, as shown in Figure-1/2 in the **general response PDF**, there is a clear Pareto Front Collapse phenomenon for all low-resource directions. And the double power law can fit well (r^2=0.86 for low-resources, r^2=0.92 for high-resources, r^2 closer to 1 suggests a better fit) under different data-adequacy/language conditions. The results confirm that the Pareto Front Collapse phenomenon and the double power law hold for both X-to-en and en-to-X directions across these diverse languages, further strengthening our claim of the power law's effectiveness in multilingual machine translation with more diverse languages and directions.
>
> > W3: When can one rely on an already instantiated DPL or, conversely, when do the parameters of the DPL need to be re-estimated?  How many times the model needs to be retrained to re-estimate the parameters of DPL in different cases?
> >
>
> We can rely on the instantiated DPL when the model and the training hyper-parameters (e.g., batch size, total update steps …) are not changed. The DPL is robust when introducing more languages or changing the translation directions, which is shown in the previous answer. When changing from EN-X to X-EN directions, we would get a better fitting result if we simply re-estimating (by retraining the model for 4 times by changing one direction’s sampling ratio from 0.1 to 0.9 equally) the $\gamma$ term of DPL (r^2: 0.86 → 0.91 for low-resources,  r^2=0.92 → 0.97 for high-resources), leaving other terms fixed.  When changing the model size, all parameters need to be re-estimated with the method listed in Section 3.2 of the original paper. It costs about 15 times to retrain the model with different ration combinations. In fact, when the estimation of DPL is done, we don’t need to re-estimate the parameters of DPL when adding new languages to the model as long as the model size is not changed.
>
> > W4: The training time recorded in Table 2 does not factor in time spent on hyperparameter searching, such as temperature-based sampling or DPL parameter estimation. Therefore, clarification on the above two points is crucial.
> >
>
> **Once** the estimation of DPL is done, we did not re-estimate it again in the experiment section. The same DPL is transferred to the 2-direction and the 4-direction setting. While for temperature-based sampling, we need to conduct the search whenever the number of directions or languages changes. We would clarify it in the revised manuscript.
>
> > Q1: Why Figure 2 and Figure 3 describe Pareto front collapse using different indicators, BLEU, and cross-entropy.
> >
>
> We use different indicators to show that the Pareto front collapse phenomenon is consistent under different evaluation metrics. We would clarify it in the revised manuscript.

---

> > ### Comment · Reviewer_hKkr · 2023-08-21
> >
> > >  W1: ... we have integrated the model-based metric, BERTScore ...
> >
> > I have never recommended the use of BERTScore as an evaluation metric for translation. Note that I recommend using COMET and BLEURT. In fact, BERTScore agrees poorly with human evaluations [1]. If you insist on using these outdated metrics, then your improvements are questionable.
> >
> > > W2: ... we have included X-EN directions and more languages ...
> >
> > What about non-English-centric MNMT (the most important thing)? In fact, English-centric MNMT has been validated to suffer from serious off-target problems, which can be greatly alleviated by non-English-centric MNMT [2]. Moreover, the development of MNMT has shifted to non-English-centric [3,4], and it is not clear whether DPL will work in this context.
> >
> > > W3: ...
> >
> > Ditto. What about non-English-centric MNMT?
> >
> >
> >
> > Given these concerns, I would lower the rating.
> >
> >
> >
> > [1] Results of WMT22 Metrics Shared Task: Stop Using BLEU – Neural Metrics Are Better and More Robust (Freitag et al., WMT 2022)
> >
> > [2] Understanding and Mitigating the Uncertainty in Zero-Shot Translation (Wang et al., arXiv 2022)
> >
> > [3] No language left behind: Scaling human-centered machine translation (NLLB Team, arXiv 2022)
> >
> > [4] Beyond english-centric multilingual machine translation (Fan et al., JMLR 2021)

---

> > > ### Author Response · Authors · 2023-08-21
> > > **Reply to reviewer hKkr's response**
> > >
> > > Dear Reviewer hKkr,
> > >
> > > Thank you for your response and advice. We would address your concerns as follows:
> > >
> > > > W1: About evaluation metrics.
> > > >
> > >
> > > We have incorporated the COMET (Unbabel/wmt22-comet-da) score into our evaluation process, which is the most human-aligned metric according to [1]. The results indicate that the Pareto Front Collapse phenomenon is consistent with the patterns observed for BERTScore, BLEU scores, and cross-entropy loss.
> > >
> > > In the experimental section, the DPL method results in a COMET score improvement of 0.5 points (4-direction average of 49.7 → 50.2, p<0.05) and 1.2 points (2-direction average of 51.4 → 52.6, p<0.05) when compared to the best Temperature-based Sampling method. We would add the result to the revised paper. The detailed results are presented below:
> > >
> > > | 2-d (COMET) | DE | HI | AVG |
> > > | --- | --- | --- | --- |
> > > | T=2 (best T) | 55.4 | 47.3 | 51.4 |
> > > | DPL (Ours) | 55.6 | 49.5 | 52.6 |
> > >
> > > | 4-d (COMET) | FR | DE | ZH | HI | AVG |
> > > | --- | --- | --- | --- | --- | --- |
> > > | T=5 (best T) | 54.4 | 52.4 | 46.8 | 45.2 | 49.7 |
> > > | DPL (Ours) | 53.9 | 52.1 | 48.2 | 46.5 | 50.2 |
> > >
> > > > W2 & W3: About non-English-centric MNMT.
> > > >
> > >
> > > We have collected wmt19-fr-de (5M), wmt19-fr-de (300K), wmt19-de-fr (5M), and wmt19-de-fr (300K) data and incorporated these additional non-English-centric directions into our MNMT experiments.  Under low-resource settings, the wmt19-fr-de (300K) and wmt19-de-fr (300K) directions exhibit a similar Pareto Front Collapse phenomenon as other low-resource English-centric directions, where the validation loss first decreases and then increases with the rise in sampling ratio. Under high-resource settings, the validation loss of wmt19-fr-de (5M) and wmt19-de-fr (5M) consistently decreases as the sampling ratio increases, which is in line with other high-resource directions. DPL also demonstrates robustness under such non-English-centric MNMT settings (r^2=0.79 for low-resource directions, r^2=0.90 for high-resource directions). The pdf in the general response can not be updated, we would add the figure of the performance changing trend to the revised paper, which has similar trends as those in current pdf.
> > >
> > > We believe the Pareto Front Collapse phenomenon would be more pronounced in non-English-centric MNMT due to the limited training data for most non-English-centric directions at current stage. Simply upsampling these directions results in a Pareto Front Collapse similar to that observed in low-resource English-centric directions.  Due to time limit, we could only experiment on those 4 directions, we would continue experimenting on more non-English-centric directions and update the results.
> > >
> > > Thank you for your efforts, and we appreciate your feedback in making this work more solid.
> > >
> > > [1] Results of WMT22 Metrics Shared Task: Stop Using BLEU – Neural Metrics Are Better and More Robust (Freitag et al., WMT 2022)

---

> > > > ### Comment · Reviewer_hKkr · 2023-08-21
> > > >
> > > > Glad to see these. While the results can't be updated, I'd like to believe it. I will raise the score accordingly.

---

### Official Review · Reviewer_wMmE · 2023-07-06

**Soundness:** 3 good
**Presentation:** 2 fair
**Contribution:** 3 good
**Rating:** 6
**Confidence:** 4

**Summary:**

This paper investigates the multi-task scaling laws for multilingual machine translation (mNMT) in the presence of *limited data* for at least one of the tasks/language pairs.

Extending Fernandes et. al. (2022) work on the data-rich scenario, the authors found that their proposed multi-task scaling law doesn't hold when data for different languages pairs is imbalanced: in this unbalanced scenario, there is a collapse of the *Pareto frontier*: increasing the weight of one LP doesn't not necessarily increase the performance of that model in that LP.

To address this, they introduced a novel *Double Power Law* (DPL) that introduces an extra term to Fernandes et. al. (2022)'s law that takes into account *overfitting*. Based on this DPL, they also porpose a method for finding the optimal weighting of the LPs, showing that it outperforms other multi-task optimizers according to BLEU.

**Strengths:**

of your tasks is low-resource. While this paper focus on mNMT, this is aksi the case (implicitly) for most LLMs trained today.

Prior work by Fernandes et. al. (2023) characterized the trade-off in the case where all tasks have pratically unlimited data, and introduced a multi-task scaling law for balancing tasks/LPs in this case. However, as the authors point out, in real life most tasks (language pairs in MT) have very limited data, and so the introduced DPL scaling law has the potential to have a lot of practical impact in balancing languages/tasks during the training of models. Their analysis relating sharpness and overfitting to the collapse of the pareto frontier is also quite interesting.

**Weaknesses:**

Despite the potential impact of the this work, I think it is quite limited by its small analysis of impact of *data* term: they introduce powerlaw term for data and add an overfitting term. However the proposed Double Power Law is given very little theoretical motivation, and little empirical validation (with non of the standard fitting metrics like r2 reported). This is exharcebated by the fact that the data terms are fitted with two or three different possible data sizes, and its hard to trust the scaling law without some validations.

Nevertheless the current findings are already quite prevalent, but the analysis is further limited by the fact they only explored one “direction” of multiliguality: and language pairs are *from* English into XXX. Fernandes et. al. (2023) also explored the opposite XX→EN direction, and found different behavior in the Pareto curves, so understanding what happens for LR in that direction is quite pertinent.

**Questions:**

- In the DPL equation (5), the “capacity occupatio ratio” seems to the same as the “effective parameter ratio” in Fernandes et al. (2023), but its functional form is different (for example, the main on in the paper had 3 fittable parameters). Why did you change it?

**Limitations:**

See above

---

> ### Author Rebuttal · Authors · 2023-08-09
>
> Dear reviewer, we really appreciate your effort in reviewing our paper, and thank you for your helpful comments. We are glad that you recognize our work 1) having the potential to have a lot of practical impact in balancing languages/tasks during the training of models 2) analysis relating sharpness and overfitting quite interesting. We address your questions below:
>
> > W1: Despite the potential impact of this work, I think it is quite limited by its small analysis of the impact of the *data* term: they introduce a powerlaw term for data and add an overfitting term. However, the proposed Double Power Law is given very little theoretical motivation, and little empirical validation (with non of the standard fitting metrics like r2 reported).
> >
>
> We have added the result of the r^2 evaluation to our fitting results. As shown in Figure1/2  in the **general response PDF,** the double power law can fit well (r^2=0.86 for low-resources, r^2=0.92 for high-resources, r^2 closer to 1 suggests a better fit) under different data-adequacy/language conditions.
>
> > W2: Nevertheless the current findings are already quite prevalent, but the analysis is further limited by the fact they only explored one “direction” of multiliguality: and language pairs are *from* English into XXX. Fernandes et. al. (2023) also explored the opposite XX→EN direction, and found different behavior in the Pareto curves, so understanding what happens for LR in that direction is quite pertinent.
> >
>
>
> We conducted additional experiments on X-EN directions with more diverse languages. The conclusion is the same as EN-X directions that there also exists Pareto Front Collapse phenomenon in low-resource directions and the double power law can well fit the results. Please refer to the general response for more details.
>
> As for the difference, we find that the pareto front collapse phenomenon is more severe for X-EN directions than EN-X directions. Fixing all the parameters of DPL except for the $\gamma$ term, increasing it to -0.25 would result in the best fit for X-EN directions (r^2: 0.86 → 0.91 for low-resources,  r^2=0.92 → 0.97 for high-resources). It means the Intrinsic Over-fitting term has more impact on the final performance in X-EN directions. We will add the discussion to the revised manuscript.
>
> > Q1: In the DPL equation (5), the “capacity occupation ratio” seems to the same as the “effective parameter ratio” in Fernandes et al. (2023), but its functional form is different (for example, the main on in the paper had 3 fittable parameters). Why did you change it?
> >
>
> We think there might be some misunderstanding about DPL equation (5), the “capacity occupation ratio” **has the same functional form** as the “effective parameter ratio” in Fernandes et al. (2023), which has only **two** fittable parameters $k$ and $\alpha$ while $p$  is the input variable.

---

> > ### Comment · Reviewer_wMmE · 2023-08-18
> > **Response to Rebuttal**
> >
> > I would like to thank the authors for their rebuttal response, and the additional experiments. I think they add alot of value to the paper!
> >
> > In particular the reverse direction experiments bring some clarity weather the DPL generalizes (at least across MT). It would also be nice to have some visualizations in terms of *trade-off* frontienrs (similar to the ones in Fernandes et (2023)).
> >
> > Additional the BLEURT experiments are nice, but a future version should include full scaling law derivations and prediction evaluation using BLEURT as a loss (also similar to Fernandes et al. (2023)).
> >
> > >  DPL equation (...)  has the same functional form as (...) Fernandes et al. (2023), which has only two fittable parameters (...)
> >
> > Hmm, equation 12 in clearly shows that the ratio approximator adds *three* fittable parameters to the scaling law (c1, c2, c3), plus the others already existing (such as alpha). Maybe the authors are instead referencing the simpler ratio approximator in the appendix (equation 13), which only adds one parameter (c1, is seems to be roughly equivalent to k)?
> >
> > Neverthless, I think the paper is now stronger, and I've revised my score.

---

> > > ### Author Response · Authors · 2023-08-20
> > > **Thanks for your helpful comments!**
> > >
> > > We are pleased to see that we have addressed your concerns and greatly appreciate all your valuable suggestions!
> > >
> > > > DPL equation (...) has the same functional form as (...) Fernandes et al. (2023), which has only two fittable parameters (...)
> > >
> > > In our rebuttal, we referred to the basic single-language-pair model's scaling law (equation 2 in Fernandes et al.). The Capacity Occupation term is derived from this simple single-language-pair scaling law, which also has 2 parameters.
> > >
> > > Thank you once again for your insightful feedback, which has significantly contributed to enhancing the quality and rigor of our paper!

---

### Official Review · Reviewer_mnLR · 2023-07-07

**Soundness:** 2 fair
**Presentation:** 3 good
**Contribution:** 3 good
**Rating:** 6
**Confidence:** 4

**Summary:**

This paper proposes to model the scaling law of multilingual machine translation using the double power law, which considers both the sampling ratio of a language and the size of the corpus. The motivation is that simply increasing the sampling ratio of low-resource language might lead to decreased performance for both high and low-resource languages due to overfitting. The paper then uses the estimated formula to improve multilingual NMT that jointly trains on two or three languages.

**Strengths:**

1. The paper is well written and it includes a good motivation and analysis section on why we should consider dataset size in the scaling law estimation
2. The resulting method is compared to strong baselines on multilingual NMT system for 2 to 4 languages. The method does seem to bring some improvement over the baseline.

**Weaknesses:**

1. The experiment is restricted to maximum of 4 languages, which is rather limited. It would be more helpful and convincing to the readers if there are multilingual NMT experiment for 10+ languages
2. The paper focuses on addressing the overfitting issue with low-resource languages, but the experiment only includes Hindi and Chinese as the “low-resource” languages.

**Questions:**

Have you considered using more true low-resource languages, such as some languages listed here: https://www.statmt.org/wmt21/large-scale-multilingual-translation-task.html

---

> ### Author Rebuttal · Authors · 2023-08-09
>
> Dear reviewer, we really appreciate your effort in reviewing our paper, and thank you for your helpful comments. We are glad that you recognize our analysis of Pareto front well motivated and our method effective. We address your questions below:
>
> > W1/W2/Q1: The experiment is restricted to maximum of 4 languages, which is rather limited. It would be more helpful and convincing to the readers if there are multilingual NMT experiment for 10+ languages.  / The paper focuses on addressing the overfitting issue with low-resource languages, but the experiment only includes Hindi and Chinese as the “low-resource” languages.  / Have you considered using more true low-resource languages …
> >
>
> We have expanded our analysis by conducting additional experiments that encompass 10 language languages such as French (FR), German (DE), Estonian (ET), Finnish (FI), Romanian (RO), Turkish (TR), Gujarati (GU), Hindi (HI), Chinese (ZH), and Latvian (LV). The dataset description is listed in Table 1 from  the **general response PDF.**
>
> As illustrated in Figures 1 and 2 in the **general response PDF**, a clear Pareto Front Collapse phenomenon is evident for all low-resource language directions in the X-EN direction. Moreover, the double power law demonstrates a strong fit (r^2=0.86 for low-resource languages and r^2=0.92 for high-resource languages, with r^2 values closer to 1 indicating a better fit) across various data-adequacy and language conditions. The direction that has the least data(gu-en,80k) shows the strongest collapse phenomenon.
>
> These findings substantiate the presence of the Pareto Front Collapse phenomenon and the effectiveness of double power law in both X-to-EN and EN-to-X language directions across a diverse range of languages. .

---

> > ### Author Response · Authors · 2023-08-21
> > **Looking Forward to Your Response**
> >
> > Dear Reviewer mnLR,
> >
> > We believe that our response has addressed your concerns regarding experiments with more diverse languages. As the discussion phase ends soon, we would greatly appreciate a brief response to confirm whether your concerns have been resolved. Please know that we are always open to further discussion.
> >
> > Thank you for your time and consideration.

---

> > > ### Comment · Reviewer_mnLR · 2023-08-22
> > >
> > > Thanks for adding the additional experiments. I will raise the score accordingly.

---

### Official Review · Reviewer_4R4K · 2023-07-08

**Soundness:** 3 good
**Presentation:** 3 good
**Contribution:** 3 good
**Rating:** 7
**Confidence:** 4

**Summary:**

This work proposes a power law for multilingual machine translation, a magic formula and constants to predict the performance of multilingual machine translation, that trades the model capacity and generalization based on the number of parameters and the training data sizes. The law is derived from the observation that the Pareto front, the performance curve by treating each language direction as an objective, collapses when the training data is highly imbalanced. They hypothesize that the root cause is the trade of the risk of overfitting when upsampling low resource, and thus, derived the power low. Experimental results show that the simple law achieved better results when compared with other competitive methods, e.g., simply tweaking temperature parameter to affect sampling ration and/or gradient manipulation-based methods.

**Strengths:**

* The analysis of Pareto front is very interesting by treating each language direction as an objective, and it clearly shows the collapse when training data size is largely skewed.

* The derived power law is very simple, yet backed by the observation of the Pareto front analysis.

* Extensive analysis on English-to-{French,German,Chinese,Hindi} directions and the empirical results show the effectiveness of the proposed power law.

**Weaknesses:**

* It is experimented only for en-to-X directions, and not X-to-en directions. I'd like to know if the law is also applicable to those directions, in addition to X-to-X directions.

* Similarly, it would be nice to investigate zero-shot directions by focusing on language families to see if transferability will also follow similar trends.

* Translation qualities are measured only by BLEU. It would be good to add different metrics, e.g., COMET or BERTScore, to see if similar trends are happening to those metrics.

**Questions:**

* Does the double power law hold for X-to-en directions? How about X-to-X directions in general?

**Limitations:**

This wok needs discussion on the applicability to a model with more diverse languages.

---

> ### Author Rebuttal · Authors · 2023-08-09
>
> Dear reviewer, we really appreciate your effort in reviewing our paper, and thank you for your helpful comments. We are glad that you recognize our analysis of Pareto front very interesting, the simplicity of the derived power law and the effectiveness of its application. We address your questions below:
>
> > W1/Q1/L1: It is experimented only for en-to-X directions, and not X-to-en directions. I'd like to know if the law is also applicable to those directions, in addition to X-to-X directions.             Does the double power law hold for X-to-en directions? How about X-to-X directions in general?            This wok needs discussion on the applicability to a model with more diverse languages.
> >
>
> We appreciate your interest in the robustness of our findings and would like to address your concerns. We added additional experiments covering 10 diverse languages:  French (FR), German (DE), Estonian (ET), Finnish (FI), Romanian (RO), Turkish (TR), Gujarati (GU), Hindi (HI), Chinese (ZH), and Latvian (LV). The dataset description is listed in Table 1 from the **general response PDF.**
>
>  For X-EN directions, as shown in Figure 1 and 2 in the **general response PDF**, there is clear Pareto Front Collapse phenomenon for all low-resource directions. And the double power law can fit well (i.e., r^2=0.86 for low-resources and r^2=0.92 for high-resources) under different data-adequacy/language conditions. The results confirm that Pareto Front Collapse phenomenon and the double power law hold for both X-to-en and en-to-X directions across these diverse languages, further strengthening our claim of the power law's effectiveness in multilingual machine translation with more diverse languages and directions.
>
> > W2: Similarly, it would be nice to investigate zero-shot directions by focusing on language families to see if transferability will also follow similar trends.
> >
>
> We conducted zero-shot experiments on the MNMT model trained on 10 from-EN and 10 to-EN directions. We discovered that in the zero-shot experiment, there are significant off-target issues, such as sentences being translated into incorrect languages or not being translated at all. These issues make it challenging to obtain a clear comparison of translation quality under such conditions. Even zero-shot direction among high-resource languages would suffer (e.g., fr→de 36% off-target ratio). We would discuss about it in the revised manuscript.
>
> > W3: Translation qualities are measured only by BLEU. It would be good to add different metrics, e.g., COMET or BERTScore, to see if similar trends are happening to those metrics.
> >
>
> We recognize that relying exclusively on BLEU scores may not offer a comprehensive understanding of translation quality. In response to your suggestion, we have incorporated model-based metric, BERTScore(https://github.com/Tiiiger/bert_score), into our evaluation. The results demonstrate that the Pareto Front Collapse phenomenon and the trends in BERTScore changes are consistent with those observed for BLEU scores and cross-entropy loss. Please refer to the general response for the detailed BERTScore results.

---

> > ### Comment · Reviewer_4R4K · 2023-08-18
> > **Read your rebuttal**
> >
> > Thanks for your additional inputs.
> >
> > The findings are quite interesting to me and I feel is this work is a nice contribution to MT.

---

> > > ### Author Response · Authors · 2023-08-18
> > > **Thanks for your encouraging comments!**
> > >
> > > It is a great encouragement to our efforts, and we are happy to see that we have addressed your concerns. We appreciate all your suggestions!

---

### Author Rebuttal · Authors · 2023-08-09

# General Response for all Reviewers

Dear Reviewers and ACs,

We sincerely appreciate the reviewers' effort in reviewing our paper and the AC for reviewing and organizing discussions! We are encouraged to hear that our analysis of Pareto front in MNMT very interesting (R1,R3), well motivated(R2), insightful(R4) and addressing an important problem(R5)

We have submitted specific responses that address each reviewer's concerns and questions. We also added a **general response PDF** which includes the results for the added experiments. In addition, we would like to highlight some shared points that arise in the reviews:

1. Whether the Pareto Front Collapse phenomenon exsits in the X-EN directions and more diverse languages.

The answer is yes! We conducted additional experiments covering 10 diverse languages in X-EN directions: French (FR), German (DE), Estonian (ET), Finnish (FI), Romanian (RO), Turkish (TR), Gujarati (GU), Hindi (HI), Chinese (ZH), and Latvian (LV). The dataset description is listed in Table 1 from the **general response PDF.**  For X-EN directions, as shown in Figure 1 and 2 in the **general response PDF**, there is clear Pareto Front Collapse phenomenon for all low-resource directions. And the double power law can fit well (i.e., r^2=0.86 for low-resources and r^2=0.92 for high-resources) under different data-adequacy/language conditions.

2. Model-based evaluation metric.

We realize that relying solely on BLEU scores may not fully capture translation quality. Thus, we've included BERTScore in our evaluation. The results show consistent trends with BLEU scores and cross-entropy loss regarding the Pareto Front Collapse phenomenon. In our experiments (Table 2 in the original paper), the DPL method leads to a 0.4-point (4-direction avg: 70.4 → 70.8, p<0.05) and 1.0-point (2-direction avg: 75.4 → 76.4, p<0.05) BERTScore improvement compared to the best Temperature-based Sampling method.

The detailed result for BERTScore is listed below:

| 2-d (bert-score) | DE | HI | AVG |
| --- | --- | --- | --- |
| T=2 (best T) | 78.5 | 72.3 | 75.4 |
| DPL(Ours) | 78.2 | 74.6 | 76.4 |

| 4-d (bert-score) | FR | DE | ZH | HI | AVG |
| --- | --- | --- | --- | --- | --- |
| T=5 (best T) | 71.2 | 74.4 | 70.5 | 65.5 | 70.4 |
| DPL(Ours) | 71 | 74.1 | 71.6 | 66.3 | 70.8 |

We hope to get valuable comments from the reviewers, and we are ready and glad to engage in active discussions.

**Please download the PDF file in this response to see the figures of the X-EN experiments' result.**

---

> ### Comment · Reviewer_hKkr · 2023-08-18
>
> Acknowledging that I've read the response.

---

### Decision · Program_Chairs · 2023-09-21

**Decision:**

Accept (poster)

**Comment:**

Paper investigates multi-task scaling laws for multilingual neural machine translation by extending Fernandes et al. 2022 and proposes a new parametrization of the scaling laws, called Double Power Law, that is able to predict out of sample points in the multilingual translation experiments. By using the proposed scaling law, authors report improved quality on multilingual 2-English and out-of-English translation scenarios covering at most 10 languages, measure both automatic (BLEU) and model based metrics (COMET, BERTscore).

Reviewers raised issues in regards to experimental design (number of languages, directionality of translation), evaluation protocols (number of languages, lack of model based metrics) and formulation/parametrization of the scaling law. Authors addressed all the issues raised by reviewers hence reviewers updated their scores.

With the current form, paper is adding practical value to the multilingual translation, and llm training in multilingual setup with its updated scaling law (DPL). Accept as a poster.